# Functional Profile of Older Adults Hospitalized in Convalescence Units of the National Network of Integrated Continuous Care of Portugal: A Longitudinal Study

**DOI:** 10.3390/jpm11121350

**Published:** 2021-12-11

**Authors:** Ana Ramos, César Fonseca, Lara Pinho, Manuel Lopes, Henrique Oliveira, Adriana Henriques

**Affiliations:** 1Nursing Research, Innovation and Development Centre of Lisbon (CIDNUR), Nursing School of Lisbon (ESEL), 1600-096 Lisbon, Portugal; ramos.anafilipa@gmail.com (A.R.); ahenriques@esel.pt (A.H.); 2Escola Superior de Enfermagem de São João de Deus, Universidade de Évora, 7000-801 Evora, Portugal; cfonseca@uevora.pt (C.F.); mjl@uevora.pt (M.L.); 3Comprehensive Health Research Centre (CHRC), Universidade de Évora, 7000-801 Evora, Portugal; 4Instituto de Telecomunicações, 1049-001 Lisbon, Portugal; hjmo@lx.it.pt; 5Instituto de Saúde Ambiental (ISAMB), Faculdade de Medicina, Universidade de Lisboa, 1649-028 Lisbon, Portugal

**Keywords:** older adults, functional status, health care, hospitalization, activities of daily living

## Abstract

Aim: To evaluate the evolution of the functional profile of older adults admitted to a health unit in Portugal; to relate the functional profile of these individuals with age, sex, education level and emotional state; and to evaluate the probability of the degree of dependence as a function of age and sex. Methods: longitudinal, retrospective study with a sample of 59,013 older adults admitted to convalescence units of the National Network of Integrated Continuous Care of Portugal. Results: In the first 75 days of hospitalization, activities of daily living, mobility and cognitive state improved, but there was a decline after 75 days of hospitalization. The ability to perform instrumental activities of daily living improved in the first 15 days of hospitalization, stabilized until 45 days and then began to worsen. Women had a higher probability of having a severe/complete dependence three years earlier than men (88 years to 91 years). A higher education level and stable emotional state were protective factors against functional decline. Conclusions: The functional profile of older adults improved during the length of stay recommended for hospitalization in convalescence units (30 days). It is critical for health systems to adopt strategies to prevent declines in the emotional state of frail individuals.

## 1. Introduction

With the increasing aging of the population and the increase in life expectancy, it is necessary to pay greater attention to the health of older adults. The European Pathway Association states that for care pathways to be successful, they must obey a set of principles, among which the following stand out: (a) the definition of clear care goals, based on scientific evidence, the best clinical practices and the expectations and characteristics of the person being cared for; (b) the facilitation of communication among all those involved; (c) effective coordination; (d) the correct monitoring and evaluation of results; and (e) the identification of the appropriate resources for the individual and the clinical situation [1]. In this sense, a care pathway should be based on both the integration of care and the continuity of care [2].

The integration of care is considered by the World Health Organization (WHO) to be the result of multifaceted efforts made to promote integration, with benefits for people [3]. It has been considered an international priority in health policy and health management research [4]. Recent data confirm that integrated care models have had benefits in improving the health-related quality of life and functionality of people with multimorbidity and frailty [5]; in reducing hospitalization and readmission rates [6]; in reducing polypharmacy [7]; and in improving patient satisfaction, perceived quality of care and access to services [8].

In Portugal, the National Network of Integrated Continuous Care (RNCCI, acronym in Portuguese) was created in 2006 through Decree Law no. 101/2006, which is intended for people who, regardless of age, are functionally dependent. The RNCCI was developed as an integrated Health and Social Security response that mobilizes the public, private and social sectors. Its intervention objectives are the rehabilitation, readaptation and reintegration of frail individuals who no longer require acute hospital care [9]. The RNCCI is focused on community outreach services and includes hospitals, health centers, district and local social security services, the Solidarity Network and local authorities. There are several types of care provided in the RNCCI, which includes inpatient units (convalescence units; medium-term and rehabilitation units; long-term and maintenance units; and level 1 pediatric integrated inpatient units) and outpatient units (mental health integrated continuous care units; day and autonomy promotion units; pediatric outpatient units and integrated continuous care teams) [9].

Convalescence Units (CUs) are intended for individuals with a potentially recoverable transitory loss of autonomy, and their purpose is clinical stabilization and functional rehabilitation. These units have their own facilities and are connected to an acute care hospital to ensure health care 24 h a day over a total length of hospitalization of 30 days [9].

As the purpose of CUs is functional rehabilitation, it seems clear to us the importance of assessing the functionality of patients to better evaluate health outcomes. The WHO developed the International Classification of Functioning, Disability and Health (ICF) to standardize the international assessment of functioning and disabilities related to the health-disease process, taking into account the body’s structures and functions and environmental factors [10]. Studies indicate that the evaluation of functionality is crucial in care models for older adults [11,12].

The performance of self-care behavior is considered extremely important for the individual person and for the health system, due to the benefits associated with it [13]. The gains from the development of self-care health behavior are related to reduced risk of complications, healthcare expenditures, hospital readmission rates, increased satisfaction with care, feelings of responsibility, control, independence, and autonomy, adoption of effective coping strategies, improved well-being, functional capacity, quality of life, symptom control, and pain [14,15,16]. In the care process, we seek to assess dependence in self-care skills (mobility, basic and instrumental activities of daily living, and cognitive status) [17,18], with the purpose of maintaining life, healthy functioning and personal development. The self-care deficit presents itself in degradé, as it can fluctuate in different levels, from mild to complete/severe [15]. The WHO report for the 2021–2030 decade emphasizes the importance of implementing actions that improve the functional ability of older adults, presenting four areas of action for this purpose: a) changing the way we think, feel and act toward age and aging; b) ensuring that communities foster the abilities of older people; c) providing integrated person-centered care and services that meet the needs of older people; and d) providing access to long-term care for older people who need it [19]. One of the strategies defined in the same report to accelerate the implementation of functional ability is to strengthen data, research and innovation [19]. Thus, the aim of the present study is to assess the functional trajectory of older adults hospitalized in CUs of the RNCCI of Portugal. For this purpose, the following objectives were defined: (1) to evaluate the evolution of the functional profile of older adults hospitalized in CUs; (2) to relate the functional profile of these individuals with age, sex, education level and emotional state; and (3) to evaluate the probability of the degree of dependence as a function of age and sex.

## 2. Materials and Methods

### 2.1. Study Type and Sample

This was a longitudinal, retrospective study with a sample of 59,013 older adults aged 65 or older hospitalized in health units belonging to CUs of the RNCCI of Portugal.

### 2.2. Instrument

To evaluate the functional profile, variables of the International Classification of Functioning, Disability and Health (ICF), which contains the ICF components: Body Functions (Mental Functions) and Activities and Participation (Mobility, Self-care, Communication, Domestic Life, Main life areas), were used. The ICF items were transformed into a Likert scale so that they could be analyzed (no problem = 1; mild or moderate problem = 2; severe problem = 3; complete problem = 4). The global Cronbach’s α = 0.951 is obtained, which means excellent internal consistency.

### 2.3. Data Collection

The data were collected from the records made by health workers and entered in the RNCCI portal. To trace the sociodemographic profile, data related to the first evaluation of each hospitalization episode occurring between 1 January 2010, and 27 February 2017, were selected.

Subsequently, the evolution of the functional profile was evaluated through the analysis of biweekly evaluations of older adults hospitalized in CUs who were targets of a set of structured professional interventions (objective 1).

To meet the second objective, which was to relate the functional profile with the variables age, sex, education level and emotional state, principal component analysis of each of the instrument domains (mobility, ADL, IADL, and cognitive state), was combined with cluster analysis. This procedure presented the advantage of reducing the number of input variables in the cluster analysis, thus helping to simplify the characterization of the upstream clusters [20].

To meet the third objective, the probabilities of each class as a function of age and by sex were evaluated.

### 2.4. Statistical Analysis

To analyze the four components of self-care capacity, over the days of hospitalization, a longitudinal analysis was performed, based on parametric tests (One-way ANOVA and t-Student test) (objective 1).

The exploratory analysis of clusters was performed using the hierarchical method (Analyze Classify Hierarchical Cluster). Since it is big data, it was necessary to perform a random partition of the database, to create a sub-sample with approximately 20% of the data, so that it was possible to process the information by SPSS. When obtaining the agglomeration coefficients (Ward’s method), a graphic projection was performed, of the highest (last 30), to visualize their distances, where it was possible to verify that the best solution resided in the retention of 3 clusters. Consecutively, the cluster analysis was carried out using the non-hierarchical optimization method available in IBM SPSS: K-means (objetive 2).

Ordinal regression with the probit link function was used to assess whether age and sex had a significant effect on the probabilities related to the type of dependence (objective 3). The link function was chosen based on the frequency distribution criteria of the classes of the dependent variable “degree of dependence” [21].

### 2.5. Ethical Procedures

The study was conducted in accordance with the guidelines of the Declaration of Helsinki and was approved by the Ethics Committee of Scientific Research in the Areas of Human Health and Welfare of the University of Évora (report number, 17036; date of approval, 26 April 2017).

## 3. Results

### 3.1. Sociodemographic and Clinical Characteristics

The mean age of the sample was 78.93 years (SD = 7.28), with an age range of 65 to 109 years. The majority of older adults admitted to CUs were in the age group of 75 to 84 years (47.2%), followed by 65 to 74 years (29.0%), and last, people aged 85 or older (23.8%). Most of the sample was female (61.5%), had a partner (44.7%), had less than 6 years of schooling (60.4%), and had an unskilled professional level at working age (70.3%). Table 1 provides the sociodemographic characterization of the sample.

### 3.2. Evolution of the Functional Profile during Hospitalization

Figure 1 shows the evolution of the functional profile of the sample throughout hospitalization, over time, with the evaluation performed every 15 days since admission. There were significant differences in each of the components: dependence in mobility (F(65.161954) = 143.337; *p* < 0.001), ADLs (F(65.166854) = 479.340; *p* < 0.001), IADLs (F(53.42791) = 9.271; *p* < 0.001) and cognitive state (F(65.161945) = 34.303; *p* < 0.001).

### 3.3. Dependence Clusters

The nonhierarchical exploratory method of cluster grouping was used, yielding the following partition: Cluster 1—29.6% (n = 11,248); Cluster 2—52.1% (n = 19,785); and Cluster 3—18.3% (n = 6932). These clusters differed significantly in the dimensions mobility (F(2.37962) = 0.162; *p* < 0.001), ADLs (F(2.37962) = 0.734; *p* < 0.001), IADLs (F(2.37962) = 0.905; *p* < 0.001) and cognitive state (F(2.37962) = 0.811; *p* < 0.001).

In general terms, the three clusters were quite distinct, with the following configuration, presented in Figure 2:(1)Cluster 1: Older adults with a higher degree of dependence (severe/complete self-care deficit);(2)Cluster 2: Older adults with an intermediate degree of dependence (moderate self-care deficit);(3)Cluster 3: Older adults with a lower degree of dependence (mild self-care deficit).

Figure 3 shows the differences among the three clusters and the variables sex, age group, education, sad/depressed emotional state and anxious emotional state.

(1)Cluster 1 (severe/complete dependence) is composed of a higher percentage of males, aged 85 years or older, older adults who did not attend school and who have been feeling depressed and anxious for a long time;(2)Cluster 2 (moderate dependence) encompasses a greater percentage of females, aged between 65 and 84 years, with 1 to 6 years of education and who have felt sad or anxious for a short time;(3)Cluster 3 (mild dependence) is predominantly composed of males aged 65 to 74 years, with more years of schooling (7 or more) and who feel depressed or anxious for a short time.

### 3.4. Degree of Dependence as a Function of Age and Sex

The assumption of the slope homogeneity model was validated ($\chi^{2}\left(2\right)=4.531$; *p* = 0.104).

The model was considered highly significant ($\chi^{2}\left(2\right)=274.822;p<0.001$), although the effect size was small ($R_{MF}^{2}=0.096;R_{N}^{2}=0.132;R_{CS}^{2}=0.129$). In the ordinal regression model, the link function “Probit” was adopted ($\Phi^{-1}\left(P\left[Y\leq k\right]\right)=\alpha_{k}-\left(0.043\times Age+0.101\times Sex_{Female}\right)$, where Φ is the standard normal distribution N(0,1)), because this function is recommended when the latent variable presents a normal distribution. The coefficients and statistical significance of the adjusted ordinal model are shown in Table 2.

The results obtained suggest that with advancing age, the probability of observing higher-order classes, i.e., the probability of observing a higher degree of dependence, increases (b_Age_ = 0.043; *p* < 0.001). Regarding sex, the results obtained suggest a higher probability of observing a higher dependence type in women than in men (b_Sex(Female)_ = 0.101; *p* < 0.001); however, this difference is moderate.

The evaluation of the probabilities of each of the classes as a function of age and by sex is shown in Figure 4. The analysis revealed that a) for both women and men, the probability of a dependence profile of “mild” is always inferior to any of the other two profile types; b) for women, starting at 88 years of age, the dependence profile that most likely occurs is “severe/complete”; for men, the same occurs, but starting at 91 years of age (three years later); c) by year of age, the ratio of the probability of observing profiles of lower dependence, compared to the probability of observing profiles of greater dependence, decreases by 4.2% (1-exp(-0.043)); and d) the odds ratio of lower dependence (mild) relative to higher dependence (severe/complete) decreases by 9.6% (1-exp(-0.101)) from males to females.

## 4. Discussion

This study evaluated the evolution of the functional profile of older adults hospitalized in CUs of the RNCCI, related it to the variables discussed below and evaluated the probability of the degree of dependence as a function of age and sex.

Regarding the evolution of self-care components throughout hospitalization, in the first 75 days of hospitalization, ADLs, mobility and cognitive state improved, but there was a decline after 75 days of hospitalization. In contrast, the ability to perform IADLs improved in the first 15 days of hospitalization, stabilized up to 45 days, and worsened thereafter. Considering that IADLs include tasks such as preparing meals, washing clothes, and housework, activities that are not performed by patients during hospitalization, it would be expected that with the lack of practice, the ability to perform these tasks would begin to decrease earlier than would ADLs, mobility or cognitive state because the tasks related to these components are maintained throughout hospitalization. However, at 75 days of hospitalization, the ability to perform these activities also begins to decline. One study reports that 30 to 60% of older adults experience a functional decline in ADL during hospitalization [22]. Low mobility during hospitalization and functional and physical changes can lead to functional deficits in ADLs and cognitive impairments [23]. However, 80% of cases are preventable with effective health care [24,25]. Importantly, individuals who are hospitalized for more than 30 days in CUs are those with less potential for rehabilitation or without family support to provide continuity of rehabilitation at home; the length of hospitalization recommended for individuals referred to CUs is 30 days, which may explain these results. Thus, it is concluded that most individuals stay for the recommended time and experience reductions in acquired deficits in all dimensions.

Regarding age, as would be expected, most individuals older than 85 years are in the severe/complete dependence cluster, the age group of 65 and 84 years is mostly in the moderate dependence cluster, and most individuals in the age group of 65 to 74 years old are in the mild dependence cluster. Other authors have also concluded that the older the age, the worse the functionality [26].

When analyzing the clusters by sex, we found that there was a higher percentage of males in the severe/complete dependence cluster and in the mild dependence cluster and a higher percentage of females in the moderate dependence cluster. However, when analyzing the probability of the degree of dependence as a function of age and sex, for women, starting at 88 years of age, the type of dependence profile that most likely occurs is “severe/complete”, while for men, the same occurs only after 91 years of age (three years later). This finding should be interpreted with caution because studies are conflicting, as some report that women have a higher degree of dependence than do men and others report that there are no differences. A literature review revealed that the incidence of functional disability was identical between sexes [27]. However, the results from two Portuguese studies with older adults indicated that the functional profile is worse in women than in men [26], regardless of age [28]. Another study that analyzed data from Spain, Portugal and Italy found that compared with men, women after 65–70 years of age had a higher risk of suffering from severe functional limitations. The same study added that in the age group of 75–80 years of age, women were 3.3% more likely than men to have severe functional limitations and that in the age group of 80 years or older, this probability increased to 15.5% [29]. The results from another study indicated that there are significant differences between sexes regarding the probability of occurrence of disability, with a higher probability for women in the following countries: United States, Korea, Southern Europe, Mexico and China. However, these differences do not exist in Northern, Central and Eastern Europe, England, and Israel. The authors conclude that gender inequality in society at the macro level is significantly associated with the probability of women developing disabilities [30]. A European study concluded that there are differences between sexes in relation to some important health indicators and that these differences are higher in Southern and Eastern Europe than in Western and Northern Europe. The same study warns that the presence of a sex difference in health cannot be considered a universal factor because, depending on the indicator and the country, the difference tended to increase, decrease or even reverse with age [31]. It should be noted that neither of the two previous studies included Portugal.

With regard to education, several studies are consistent in indicating that the lower the educational level, the worse the functional profile and the greater the degree of dependence [26,28,32] and that a lower education level is associated with worse physical and mental health outcomes [33,34,35]. In addition, a study showed that the impact of multimorbidity on ADLs was three times higher at the lowest education level than at the highest education level [36]. Another study added that after 65 years of age, the average probability of severe functional limitations for individuals with a low education level increased to more than 40% at age 80 years and older, while it was 26% in the higher education category [29]. The results of the present study are in line with those described in the literature, with a predominance of individuals who did not attend school in the severe/complete dependence cluster; those who studied for 1 to 6 years were predominant in the moderate dependence cluster, and those with a higher education level (7 or more years) were mostly in the mild dependence cluster. Thus, it is confirmed that in individuals admitted to CUs, the higher the educational level is, the lower the likelihood of dependence; additionally, education is extremely important for health and for active and healthy aging.

Regarding the analysis of emotional state, in the severe/complete dependence cluster, there was a predominance of individuals who felt depressed and anxious for a long time, and in the moderate dependence and mild dependence clusters, there was a predominance of older adults who felt sad or anxious for a short time. In fact, in a literature review, depression was considered one of the main risk factors for functional disability in older adults [27]. Another recent study that analyzed several combinations of multimorbidity concluded that those that included depression were the only consistent predictors of disability [37]. Another study followed the same direction, with the results indicating that depression and/or cognitive deficit was associated with a substantially greater potential disability than combinations composed exclusively of somatic diseases [38].

The strengths of the present study include the relatively large sample size and the heterogeneity of the participants.

The limitations of the present study include the lack of an analysis of the time period after 2017 and the inclusion of only some of the determinants identified in the literature as influencing functional ability (e.g., emotional state).

## 5. Conclusions

The recommended length of hospitalization in CUs of the RNCCI of Portugal is 30 days, although some individuals remain longer. During this period, there was a significant improvement in the functional profile of older adults hospitalized in these units, and there was a decline in those who remained beyond 75 days. More studies are needed to understand the reason for the observed decline after this cutoff point.

Women hospitalized in CUs were more likely to have a severe degree of/complete dependence three years earlier than men (88 years compared to 91 years of age, respectively). The higher the education level and the better the emotional state was, the lower the degree of dependence. This study confirms the importance of the education level and emotional state in functional ability in older adults. It is necessary that future studies evaluate the effectiveness of interventions that seek to prevent declines in the emotional state of individuals so that they are applied in clinical practice when a situation occurs that threatens the independence of such individuals, as is the case of people referred to CUs. In addition, the importance of strategies that promote literacy in the population is reinforced.

## Figures and Tables

**Figure 1 jpm-11-01350-f001:**
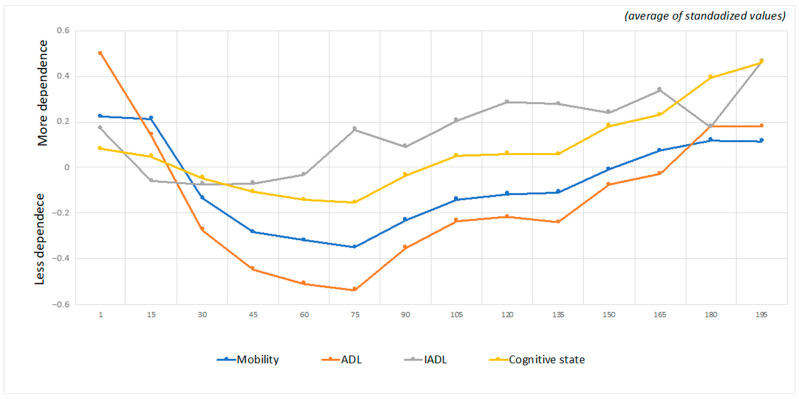
Components: mobility, activities of daily living (ADL), instrumental activities of daily living (IADL) and cognitive status in convalescence units (standardized mean values).

**Figure 2 jpm-11-01350-f002:**
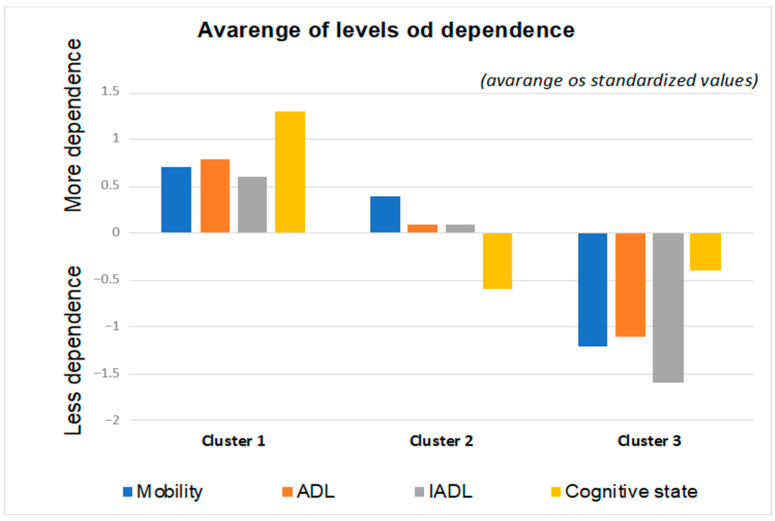
Mean dependence levels for Cluster 1, Cluster 2 and Cluster 3 in the convalescence units.

**Figure 3 jpm-11-01350-f003:**
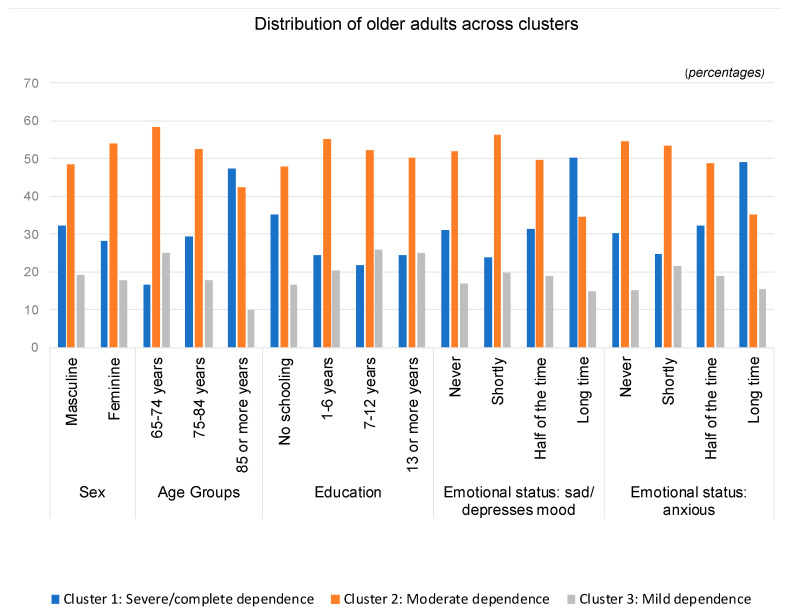
Distribution of people aged 65 years or older in the convalescence units by sex, age group, education level, emotional state: sad/depressed and emotional state: anxious, per cluster.

**Figure 4 jpm-11-01350-f004:**
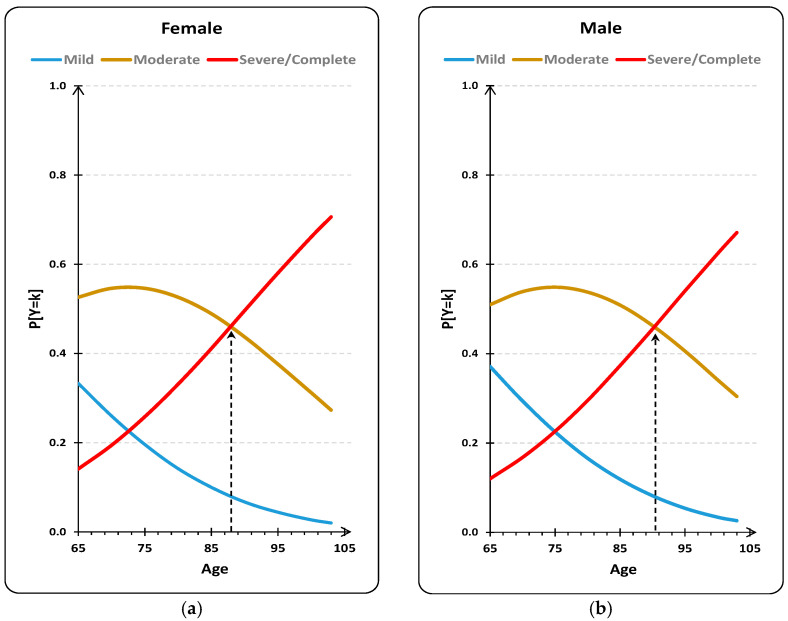
Evaluation of the probabilities for mild, moderate and severe dependence for women (**a**) and men (**b**) ($\chi^{2}\left(2\right)=274.822;p<0.001;R_{MF}^{2}=0.096;R_{N}^{2}=0.132;R_{CS}^{2}=0.129$).

**Table 1 jpm-11-01350-t001:** Sociodemographic characterization of older adults hospitalized in convalescence units by sex (2010–2017).

Sociodemographic Variables	*n* (%)
Age (years)	
65–74	15,320 (29.0)
75–84	24,987 (47.2)
≥85	12,578 (23.8)
Sex	
Female	32,535 (61.5)
Male	20,350 (38.5)
Marital status	
Single	6193 (13.2)
Married	20,763 (44.4)
Domestic partnership	454 (0.3)
Divorced	2018 (4.3)
Widowed	17,578 (37.6)
Unknown	109 (0.2)
Education (years)	
No education	8244 (31.5)
1 to 6	15,802 (60.4)
7 to 12	1087 (4.2)
≥13	1047 (4.0)
Professional Level	
Unskilled	18,379 (70.3)
Skilled	6191 (23.7)
Intermediate	1139 (4.4)
Specialist	453 (1.7)
Cohabitation	
Lives alone	7192 (27.5)
Lives with other (s)	18,988 (72.5)
Region of Portugal	
Alentejo	4913 (9.7)
Algarve	3659 (7.2)
Center	11,937 (22.6)
Lisbon and Vale do Tejo	11,362 (22.4)
North	18,805 (37.1)

**Table 2 jpm-11-01350-t002:** Estimates and significance of the adjusted “Probit” model.

Parameters		Estimate	Standard Error	$\chi_{Wald}^{2}$	df	*p* Value	95% Confidence Interval
Thresholds	Mild	α_Mild_ = 2.439	0.065	1421.020	1	<0.001	[2.312; 2.566]
	Moderate	α_Moderate_ = 3.946	0.066	3559.323	1	<0.001	[3.816; 4.075]
Localization	Age	*b_Age_* = 0.043	0.001	2684.676	1	<0.001	[0.041; 0.044]
	Sex (Female)	*b*_Sex(Female)_ = 0.101	0.012	69.659	1	<0.001	[0.078; 0.125]

## Data Availability

Data are available from the authors upon reasonable request and with permission of University of Évora.

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
