# Peer review of "Functional Profile of Older Adults Hospitalized in Convalescence Units of the National Network of Integrated Continuous Care of Portugal: A Longitudinal Study"

_jpm, 2021, doi:10.3390/jpm11121350_

Round 1

Reviewer 1 Report

Thank you for the opportunity to review the manuscript “Functional profile of older adults hospitalized in convalescence units of the National Network of Integrated Continuous Care of Portugal: a longitudinal study” (jpm-1432518).

The aim of a longitudinal, retrospective study was to evaluate the evolution of the functional profile of older adults admitted to a health unit in Portugal and to relate the functional profile of these individuals with age, sex, education level.

This is an interesting paper with worthful data; however, some parts of the study must be carefully revised.

The introduction writing was somewhat discursive, and the factual statements are not so clear. The background would be clearer and have more context with greater details about the amount of the problem. The statements should be supported by more recent literature citations. Please combine the introduction with your results. In this context, please update references and add recent literature for a better-connected discussion.

Introduction: It is particularly important to work out the goal of improving the care processes with patient-related results.

This offers the opportunity to implement indication-based treatment pathways to improve care in healthcare facilities and reduce costs. Please add relevant references for example in pain management (doi: 10.5539/gjhs.v4n2p50) and also add other references.

It is possible to deal with the degree of chronicity and the clinical pictures of the examined patients. This would improve the informative value of the study.

Author Response

Thank you very much for your comments which have helped improve the article significantly.

We have followed your suggestions and improved the introduction.

Reviewer 2 Report

This interesting study investigated the evolution of the functional profile among older adults admitted to a health unit in Portugal. Overall the manuscript is well-written. In my opinion, despite this current evaluation of the manuscript, the information reported needs some in-depth clarification.

Comments:

  1. The keywords are not inserted properly. I would suggest fixing this.
  2. Line 97 – I believe the authors intend to say – ‘To evaluate the functional profile, …’
  3. Line 100: It is unclear when the authors stated, ‘In addition to these, we have added an item: take medication’. Does it mean the item was incorporated into ICF? Why ‘take medication’ (self or assisted) is so important to be part of the functional profile? Also, heading 2.2 seems vague. It should be an Evaluation of the instrument. Why do authors want to evaluate the ICF for the functional profile? Is it a validated measure?
  4. Section 2.3: procedures is not a proper heading here. It should be data collection and outcome measure or something relevant to this.
  5. Line 116: it can’t be ‘evolution’. Please proofread thoroughly for spell checks and grammar. In line 90, it was stated correctly as ‘evaluate.’
  6. Statistical analysis should be section 2.4. Also, the first paragraph needs clarity on the listed statistical measures related to each objective. I would suggest elaborating this based on your study objectives. For example, A chi-square test/Mann-Whitney U test/Kruskal Wallis test was applied, as appropriate, to make comparisons between groups.
  7. Statistical analysis: when the authors mentioned evaluating the instrument, I wondered whether they applied any sensitivity analysis?
  8. 2. – again, it's not ‘evolution.’
  9. Section 3.2. I am a little perplexed when the authors' state the evolution of the functional profile… I just wanted to clarify if they actually mean it or is it evaluation of….?
  10. Section 3.2. what does it mean by synthetic indices? Why are the decimal places 6? What does the F represent (is it a parameter estimate?)
  11. Did the study explore functional profiles over time (trend)?
  12. How were the clusters developed?
  13. Please follow the standard approach while plotting the values (lines 186-188). it isn’t clear to the readers (throughout the manuscript)
  14. Suggest Table 2 can be a supplementary file
  15. The first half of the discussion section doesn’t have any citations. It looks like the Study results are repetitive.

Author Response

Thank you very much for your comments which have helped improve the article significantly.

We have answered your questions below.

  1. We add the keywords.
  2. Thank you. We correct this.
  3. We have enhanced this section to better explain the instrument that was used based on the ICF. We have removed "take medication". This was an oversight.
  4. Thanks. We change 2.3 to Data Collection
  5. We change this to: “To meet the third objective, the probabilities of each class as a function of age and by sex was evaluated.”
  6. Thank you for your comment. We clarify this in the manuscript.
  7. The ICF items were transformed into a Likert scale so that they could be analyzed (no problem = 1; mild or moderate problem = 2; severe problem = 3; complete problem = 4). The global Cronbach's α = 0.951 is obtained, which means excellent internal consistency.
  8. We correct this.
  9. We evaluate the evolution of the functional profile over time. That is, several evaluations were performed throughout the hospitalization. We evaluated the evolution of the functional profile throughout the various evaluations.
  10. The synthetic indices are the self-care components. Maybe this is a translation problem. We have changed it to components for better understanding. The F is part of the analysis of simple parametric variances (one-way Anova). We clarify this in statistical analyses.
  11. Yes.
  12. We clarify this in statistical analyses.
  13. Table 2 is important to justify the model.
  14. We add citations to discuss the results.

Round 2

Reviewer 1 Report

The authors improved the paper. It can be considered for publication.

Reviewer 2 Report

The authors have done a substantial amount of revision by carefully addressing the reviewers comments. The content and clarity of the manuscript have been improved radically from the previous version. I have no further comments.